# Structure-Aware Convolutional Neural Networks

**Jianlong Chang**[1,2]    **Jie Gu**[1,2]    **Lingfeng Wang**[1]    **Gaofeng Meng**[1]
**Shiming Xiang**[1,2]    **Chunhong Pan**[1]
[1]NLPR, Institute of Automation, Chinese Academy of Sciences
[2]School of Artificial Intelligence, University of Chinese Academy of Sciences
{jianlong.chang, jie.gu, lfwang, gfmeng, smxiang, chpan}@nlpr.ia.ac.cn

## Abstract

Convolutional neural networks (CNNs) are inherently subject to invariable filters that can only aggregate local inputs with the same topological structures. It causes that CNNs are allowed to manage data with Euclidean or grid-like structures (*e.g.*, images), not ones with non-Euclidean or graph structures (*e.g.*, traffic networks). To broaden the reach of CNNs, we develop *structure-aware convolution* to eliminate the invariance, yielding a unified mechanism of dealing with both Euclidean and non-Euclidean structured data. Technically, filters in the structure-aware convolution are generalized to univariate functions, which are capable of aggregating local inputs with diverse topological structures. Since infinite parameters are required to determine a univariate function, we parameterize these filters with numbered learnable parameters in the context of the function approximation theory. By replacing the classical convolution in CNNs with the structure-aware convolution, *Structure-Aware Convolutional Neural Networks* (SACNNs) are readily established. Extensive experiments on eleven datasets strongly evidence that SACNNs outperform current models on various machine learning tasks, including image classification and clustering, text categorization, skeleton-based action recognition, molecular activity detection, and taxi flow prediction.

## 1  Introduction

Convolutional neural networks (CNNs) provide an effective and efficient framework to deal with Euclidean structured data, including speeches and images. As a core module in CNNs, the convolution unit explicitly allows to share parameters among the whole spatial domains to extremely reduce the number of parameters, without sacrificing the expressive capability of networks [3]. Benefiting from such artful modeling, significant successes have been achieved in a multitude of fields, including the image classification [15, 24] and clustering [5, 6], the object detection [9, 32], and amongst others.

Although the achievements in the literature are brilliant, CNNs are still incompetent to handle non-Euclidean structured data, such as the traffic flow data on traffic networks, the relational data on social networks, and the active data on molecule structure networks. The major limitation originates from that the classical filters are invariant at each location. As a result, the filters can only be applied to aggregate local inputs with the same topological structures, not with diverse topological structures.

In order to eliminate the limitation, we develop structure-aware convolution in which a single share-able filter suffices to aggregate local inputs with diverse topological structures. For this purpose, we generalize the classical filters to univariate functions that can be effectively and efficiently parameterized under the guidance of the function approximation theory. Then, we introduce local structure representations to quantificationally encode topological structures. By modeling these representations into the generalized filters, the corresponding local inputs can be aggregated based on the generalized filters consequently. In practice, Structure-Aware Convolutional Neural Networks (SACNNs) can be readily established by replacing the classical convolution in CNNs with our structure-aware

convolution. Since all the operations in our structure-aware convolution are differentiable, SACNNs can be trained end-to-end by the standard back-propagation.

To sum up, the key contributions of this paper are:

- The structure-aware convolution is developed to establish SACNNs to uniformly deal with both Euclidean and non-Euclidean structured data, which broadens the reach of convolution.
- We introduce the learnable local structure representations, which endow SACNNs with the capability of capturing the latent structures of data in a purely data-driven way.
- By taking advantage of the function approximation theory, SACNNs can be effectively and efficiently trained with the standard back-propagation to guarantee the practicability.
- Extensive experiments demonstrate that SACNNs are superior to current models in various machine learning tasks, including classification, clustering, and regression.

## 2 Related work

### 2.1 Convolutional neural networks (CNNs)

To elevate the performance of CNNs, much research has been devoted to designing the convolution units, which can be roughly divided into two classes, *i.e.*, handcrafted and learnable ones.

Handcrafted convolution units generally derive from the professional knowledge. Primary convolution units [24, 26] present large sizes, *e.g.*, $7 \times 7$ pixels in images. To increase the nonlinearity, stacking multiple small filters (*e.g.*, $3 \times 3$ pixels) instead of using a single large filter has become a common design in CNNs [38]. To obtain larger receptive fields, the dilated convolution [41], whose receptive field size grows exponentially while the number of parameters grows linearly, is proposed. In addition, the separable convolution [7] promotes performance by integrating various filters with diverse sizes.

Among the latter, lots of efforts have been widely made to learn convolution units. By introducing additional parameters named offsets, the active convolution [19] is explored to learn the shape of convolution. To achieve dynamic offsets that vary with inputs, the deformable convolution [9] is proposed. Contrary to such modifications, some approaches have been devoted to directly capturing structures of data to improve the performance of CNNs, such as the spatial transform networks [18].

While these models have been successful on Euclidean domains, they can hardly be applied to non-Euclidean domains. In contrast, our SACNNs can be utilized on these two domains uniformly.

### 2.2 Graph convolutional neural networks (GCNNs)

Recently, there has been a growing interest in applying CNNs to non-Euclidean domains [3, 29, 31, 35]. Generally, existing methods can be summarized into two types, *i.e.*, spectral and spatial methods.

Spectral methods explore an analogical convolution operator over non-Euclidean domains on the basis of the spectral graph theory [4, 16, 27]. Relying on the eigenvectors of graph Laplacian, data with non-Euclidean structures can be filtered on the corresponding spectral domain. To enhance the efficiency and acquire spectrum-free methods without performing eigen-decomposition, polynomial-based networks are developed to execute convolution on non-Euclidean domains efficiently [10, 22].

Contrary to the spectral methods, spatial methods always analogize the convolutional strategy based on the local spatial filtering [1, 2, 30, 31, 37, 40]. The major difference between these methods lies in the intrinsic coordinate systems used for encoding local patches. Typically, the diffusion CNNs [1] encode local patches based on the random walk process on graphs, the anisotropic CNNs [2] employ an anisotropic patch-extraction method, and the geodesic CNNs [30] represent local patches in polar coordinates. In the mixture-model CNNs [31], synthetically, learnable local pseudo-coordinates are developed to parameterize local patches in a general way. Additionally, a series of spatial methods without the classical convolutional strategy have also been explored, including the message passing neural networks [12, 28, 34], and the graph attention networks [39].

In spite of considerable achievements, both spectral and spatial methods partially rely on fixed structures (*i.e.*, fixed relationship matrix) in graphs. Benefiting from the proposed structure-aware convolution, by comparison, the structures can be learned from data automatically in our SACNNs.

# 3 Structure-aware convolution

Convolution, intrinsically, is an aggregation operation between local inputs and filters. In practice, local inputs involve not only their input values but also topological structures. Accordingly, filters should be in a position to aggregate local inputs with diverse topological structures. To this end, we develop the structure-aware convolution by generalizing the filters in the classical convolution and modeling the local structure information into the generalized filters.

The filters in the classical convolution can be smoothly generalized to univariate functions. Without loss of generality and for simplicity, we elaborate such generalization with 1-Dimensional data. Given an input $\mathbf{x} \in \mathbb{R}^n$ and a filter $\mathbf{w} \in \mathbb{R}^{2m-1}$, the output at the $i$-th vertex (location) is

$$\bar{y}_i = \mathbf{w}^{\mathrm{T}} \mathbf{x}_i = \sum_{i-m<j<i+m} w_{j-i+m} \cdot x_j, \quad i \in \{1, 2, \cdots, n\}, \tag{1}$$

where $\mathbf{x}_i = [x_{i-m+1}, \cdots, x_{i+m-1}]^{\mathrm{T}}$ is the local input at the $i$-th vertex, $i-m<j<i+m$ indicates that the $j$-th vertex is a neighbor of the $i$-th vertex, $w_{j-i+m}$ and $x_j$ signify the $(j-i+m)$-th and $j$-th elements in $\mathbf{w}$ and $\mathbf{x}$, respectively. For any univariate function $f(\cdot)$, Eq. (1) can be equivalently rewritten as follows when $f(j-i+m) = w_{j-i+m}$ is always satisfied, *i.e.*,

$$\bar{y}_i = \mathbf{f}_{\mathcal{R}}^{\mathrm{T}} \mathbf{x}_i = \sum_{i-m<j<i+m} f(j-i+m) \cdot x_j, \quad i \in \{1, 2, \cdots, n\}, \tag{2}$$

where $f(\cdot)$ is called a functional filter, $\mathcal{R} = \{j-i+m \mid i-m<j<i+m\} = \{1, 2, \cdots, 2m-1\}$, and $\mathbf{f}_{\mathcal{R}} = \{f(r)|r \in \mathcal{R}\}$. Actually, $\mathcal{R}$ encodes relationships between a vertex and its neighbors. For example, $r \in \mathcal{R}$ means that the $(i-m+r)$-th vertex is the $r$-th neighbor of the $i$-th vertex. Since the relationships in $\mathcal{R}$ can reflect the structure information around a vertex, we call $\mathcal{R}$ a local structure representation. Generally, the local structure representation $\mathcal{R}$ is constant in the classical convolution, which causes that the same $\mathbf{f}_{\mathcal{R}}$ is shared at each vertex. As a result, the classical convolution solely pertains to manage data with the same local topological structures, not with diverse ones.

To handle this limitation, we introduce general local structure representations to quantificationally encode any local topological structure, and then develop structure-aware convolution by replacing the constant $\mathcal{R}$ in classical convolution with the introduced general ones. Technically, both Euclidean and non-Euclidean structured data can be represented by a graph $\mathcal{G} = (\mathcal{V}, \mathcal{E}, \mathbf{R})$, where the vertices in $\mathcal{V}$ store the values of data, the edges in $\mathcal{E}$ indicate whether two vertices are connected, and the relationship matrix $\mathbf{R}$ signifies the structure information in the graph $\mathcal{G}$. For a vertex $i \in \mathcal{V}$, the local structure representation at $i$ is encoded via the relationships with its neighbors, *i.e.*,

$$\mathcal{R}_i = \{r_{ji}|e_{ji} \in \mathcal{E}\}, \quad i \in \{1, 2, \cdots, n\}, \tag{3}$$

where $e_{ji} \in \mathcal{E}$ means that the $j$-th vertex is a neighbor of the $i$-th vertex, $r_{ji}$ is the element of $\mathbf{R}$ at $(j, i)$ and indicates the relationship from the $j$-th vertex to the $i$-th vertex. Note that $\mathcal{S} = \{\mathcal{R}_i|i \in \mathcal{V}\}$ can include the whole structure information in the graph $\mathcal{G}$ by integrating the local structure representations together. This implies that Eq. (3) is a reasonable formulation for local topological structures. Based on the introduced local structure representations, the structure-aware convolution is developed by modeling these representations into the generalized functional filters. Formally, given an input $\mathbf{x}$ embedded on the graph $\mathcal{G}$ and a functional filter $f(\cdot)$, we define the structure-aware convolution as

$$\bar{y}_i = \mathbf{f}_{\mathcal{R}_i}^{\mathrm{T}} \mathbf{x}_i = \sum_{e_{ji} \in \mathcal{E}} f(r_{ji}) \cdot x_j, \quad i \in \{1, 2, \cdots, n\}, \tag{4}$$

where $\mathbf{f}_{\mathcal{R}_i} = \{f(r_{ji})|e_{ji} \in \mathcal{E}\}$ varies with $\mathcal{R}_i$. Benefiting from this modification, the structure-aware convolution is capable of aggregating local inputs with diverse topological structures.

# 4 Structure-aware convolutional neural networks

Replacing the classical convolution in CNNs with the structure-aware convolution, SACNNs are established. Intuitively, a structure-aware convolutional layer is illustrated in Figure 1. However, two essential problems need to be tackled before training SACNNs. First, functional filters in the structure-aware convolution are univariate functions, which need infinite parameters to be determined. This implies that SACNNs can not be learned in a common way, and an effective and efficient strategy is required to learn these filters with numbered parameters. Second, local structure representations (or the relationship matrix $\mathbf{R}$) may be hardly defined in advance and thus a learning mechanism is needed. In the following, Section 4.1 and 4.2 focus on tackling these two problems, respectively.

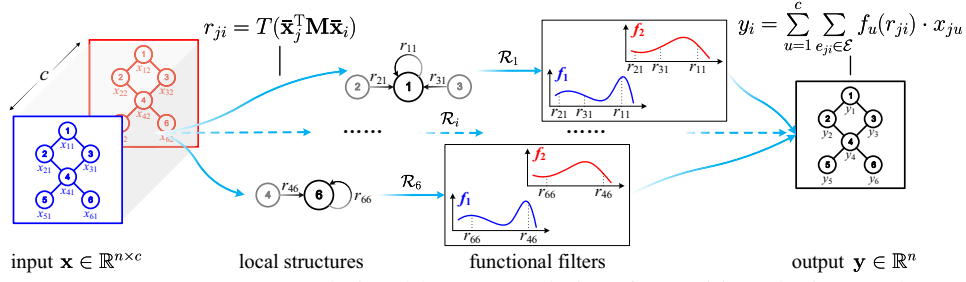

Figure 1: A structure-aware convolutional layer. For clarity of exposition, the input $\mathbf{x}$ has $c = 2$ channels with $n = 6$ vertices, the output $\mathbf{y}$ has a single channel, and $\bar{\mathbf{x}}_j$, $\bar{\mathbf{x}}_i \in \mathbb{R}^c$ indicate the $j$-th and $i$-th rows of the input $\mathbf{x}$, respectively. For each vertex $i$, its local structure representation is first captured from the input and represented as $\mathcal{R}_i$, which is identically shared for each channel of the input $\mathbf{x}$. Afterwards, the local inputs in the first and second channels are aggregated via the first filter $f_1(\cdot)$ and the second filter $f_2(\cdot)$ respectively, with the same $\mathcal{R}_i$. Note that $f_1(\cdot)$ and $f_2(\cdot)$ are shared for every location in the first and second channels, respectively.

## 4.1 Polynomial parametrization for functional filters

We parameterize the developed functional filters with numbered learnable parameters under the guidance of the function approximation theory. In mathematics, for an arbitrary univariate function $h(x)$, it can be composed of a group basis functions $\{h_1(x), h_2(x), \cdots\}$ with a set of coefficients $\{v_1, v_2, \cdots\}$, denoted by $h(x) \simeq \sum_{k=1}^{t} v_k \cdot h_k(x)$, where $h_k(x)$ and $v_k$ are the $k$-th basis function and the corresponding coefficient, respectively. The equation is satisfied when $t$ tends to infinity.

Because of the high efficiency [14], our functional filters are parameterized based on the Chebyshev polynomials that form an orthogonal basis for $L^2([-1, 1], dy/\sqrt{1-y^2})$, the Hilbert space of square integrable functions with respect to the measure $dy/\sqrt{1-y^2}$. Formally, the Chebyshev polynomial $h_k(x)$ of order $k-1$ ($k \geq 3$) can be generated by the stable recurrence relation $h_k(x) = 2xh_{k-1}(x) - h_{k-2}(x)$, with $h_1(x) = 1$ and $h_2(x) = x$. In practice, the truncated expansion of Chebyshev polynomials is employed to approximate the functional filter $f(\cdot)$ in Eq. (4), *i.e.*,

$$y_i = \sum_{e_{ji} \in \mathcal{E}} f(r_{ji}) \cdot x_j = \sum_{e_{ji} \in \mathcal{E}} \left( \sum_{k=1}^{t} v_k \cdot h_k(r_{ji}) \right) \cdot x_j, \quad i \in \{1, 2, \cdots, n\}, \tag{5}$$

where $t$ is the number of the truncated polynomials, and $\{v_1, \cdots, v_t\}$ are $t$ learnable coefficients corresponding to the polynomials $\{h_1(x), \cdots, h_t(x)\}$. Note that $f(r_{ji})$ can be cumulatively computed based on the recurrence relation, leading to an efficient computing strategy.

## 4.2 Local structure representations learning

To eliminate the feature engineering, we consider to learn local structure representations from data rather than using predefined ones. To preserve the structure consistency between channels, for every structure-aware convolutional layer, only a single local structure representation set $\mathcal{S} = \{\mathcal{R}_i | i \in \mathcal{V}\}$ is identically learned for each channel of the input. Formally, given a multi-channel input feature map $\mathbf{x} \in \mathbb{R}^{n \times c}$, where $n$ and $c$ denote the numbers of vertices and channels respectively, the local structure representation at each vertex is learned as

$$\mathcal{R}_i = \{r_{ji} = T(\bar{\mathbf{x}}_j^{\mathrm{T}} \mathbf{M} \bar{\mathbf{x}}_i) \mid e_{ji} \in \mathcal{E}\}, \quad i \in \{1, 2, \cdots, n\}, \tag{6}$$

where $\bar{\mathbf{x}}_j$, $\bar{\mathbf{x}}_i \in \mathbb{R}^c$ indicate the $j$-th and $i$-th rows of the input $\mathbf{x}$ respectively, $\mathbf{M} \in \mathbb{R}^{c \times c}$ is a matrix with $c \times c$ learnable parameters to measure relationships between local vertices, and $T(\cdot)$ is the Tanh function to normalize elements in local structure representations into $[-1, 1]$ strictly.

This local structure learning formulation has two good properties. First, $\mathbf{M}$ is identically shared for each channel of the input, so every channel possesses the same structure in each structure-aware convolutional layer and the size of $\mathbf{M}$ only depends on the number of channels. As a result, only a few additional parameters are required to be learned, which can alleviate the overfitting when training data is limited. Second, $\mathbf{M}$ is not constrained as a symmetric matrix, namely $r_{ji}$ may not be equal to $r_{ij}$. This implies that our approach is capable of modeling not only undirected structures, but also direct structures, such as the traffic networks and the social networks.

### 4.3 Understanding the structure-aware convolution

In this subsection, we give the following theorem to reveal the essence of our structure-aware convolution (the proof is reported in the supplementary material).

**Theorem 1.** *Under the Chebyshev polynomial basis, the structure-aware convolution is equivalent to*

$$y_i = \mathbf{v}^\mathrm{T} \mathbf{P}_i \mathbf{x}_i, \quad i \in \{1, 2, \cdots, n\},$$

*where* $\mathbf{v} \in \mathbb{R}^t$ *is the coefficients of the polynomials,* $\mathbf{P}_i \in \mathbb{R}^{t \times m}$ *is a matrix determined by the local structure representation* $\mathcal{R}_i$ *and the polynomials, and* $\mathbf{x}_i \in \mathbb{R}^m$ *is the local input at the* $i$*-th vertex.*

Theorem 1 indicates that the structure-aware convolution can be split into two independent units, *i.e.*, a transformation $\mathbf{P}_i \in \mathbb{R}^{t \times m}$ and a vector $\mathbf{v} \in \mathbb{R}^t$. In the first unit, the transformation $\mathbf{P}_i$ devotes to encoding the $m$-Dimensional local inputs as $t$-Dimensional vectors. Since the basis functions are fixed in the Chebyshev polynomial basis, $\mathbf{P}_i$ is purely depended on the corresponding local structure representation $\mathcal{R}_i$ that is varied with the vertex $i$ and can be learned according to Eq. (6). It is worth noting that this transformation $\mathbf{P}_i$ is similar to a specific local spatial transformer in the spatial transform networks [18]. In the second unit, the learnable vector $\mathbf{v}$ is shared by every vertex to aggregate these encoded local inputs, which is akin to the classical convolution. By integrating these two learnable units together, the structure-aware convolution can simultaneously focus on local input values and local topological structures to capture high-level representations.

## 5 Experiments

In this section, we systematically carry out extensive experiments to verify the capability of SACNNs. Due to the space restriction we report some experimental details in the supplement, such as the gradients during training, descriptions of datasets, descriptions of datasets, and network architectures. Specifically, Our core code will be released at `https://github.com/vector-1127/SACNNs`.

### 5.1 Experimental settings

We perform experiments on six Euclidean and five non-Euclidean structured datasets to verify the capability of SACNNs. Six Euclidean structured datasets include the Mnist [26], Cifar-10 [23], Cifar-100 [23], STL-10 [8], Image10 [6], and ImageDog [6] image datasets. Five non-Euclidean structured datasets contain the text categorization datasets 20NEWS and Reuters [25], the action recognition dataset NTU [36], the molecular activity dataset DPP4 [20], and the taxi flow dataset TF-198 [42] that consists of the taxis flow data at 198 traffic intersections in a city.

With respect to the scope of applications, two types of methods are compared, *i.e.*, CNNs and GCNNs. On Euclidean domains, popular CNN models, including the classical convolution (ClaCNNs) [26], the separable convolution (SepCNNs) [7], the active convolution (ActCNNs) [19], and the deformable convolution (DefCNNs) [9] are utilized for comparisons. On non-Euclidean domains, both spatial and spectral GCNNs are taken as competitors to SACNNs, including the local connected networks (LCNs) [4], the dynamic filters based networks (DFNs) [40], the edge-conditioned convolution (ECC) [37], the mixture-model networks (MoNets) [31] (which is a generalization of the diffusion CNNs [1], the anisotropic CNNs [2], and the geodesic CNNs [30]), the spectral networks (SCNs) [16], the Chebyshev based SCNs (ChebNets) [10], and the graph convolution networks (GCNs) [22]. Furthermore, SACNNs$^\dagger$ that omit the structure learning in SACNNs are used as a baseline of our method and to show the effectiveness of structure learning. In SACNNs$^\dagger$, $\mathcal{R}_i$ is assigned by uniformly sampling on $[-1, 1]$, *e.g.*, $\mathcal{R}_i = \{-\frac{1}{2}, 0, \frac{1}{2}\}$ is predefined when a 3-Dimensional filter is required.

The hyper-parameters in SACNNs are set as follows. In our experiments, the max pooling and the Graclus method [11] are employed as the pooling operations to coarsen the feature maps in SACNNs when managing Euclidean and non-Euclidean structured data respectively, the ReLU function [13] is used as the activation function, batch normalization [17] is employed to normalize the inputs of all layers, parameters are randomly initialized with a uniform distribution $U(-0.1, 0.1)$, the order of polynomials $t$ is set to the maximum number of neighbors among the whole spatial domains (*e.g.*, $t = 9$ if we attempt to learn $3 \times 3$ filters in images). During the training stage, the Adam optimizer [21] with the initial learning rate $0.001$ is utilized to train SACNNs, the mini-batch size is set to 32, the categorical cross entropy loss is used in the classification tasks, and the mean squared

Table 1: The classification or clustering accuracies on the experimental Euclidean structured datasets. For clarity, ‡ indicates that DAC [6] is used to cluster the whole samples in each experimental dataset.

| Datasets | Mnist | Cifar-10 | Cifar-100 | STL-10 | Image10‡ | ImageDog‡ | Time (s) |
|---|---|---|---|---|---|---|---|
| ClaCNNs [26] | 0.9953 | 0.9075 | 0.6629 | 0.6635 | 0.5272 | 0.2748 | **53±1** |
| SepCNNs [7] | 0.9910 | 0.9062 | 0.6643 | 0.6685 | 0.5637 | 0.2754 | 68±1 |
| ActCNNs [19] | 0.9926 | 0.9086 | 0.6648 | 0.6761 | 0.5478 | 0.2786 | 83±2 |
| DefCNNs [9] | 0.9908 | 0.8718 | 0.6349 | 0.6564 | 0.4853 | 0.2355 | 125±3 |
| SACNNs† | 0.9957 | 0.9091 | 0.6759 | 0.7175 | 0.5953 | 0.2801 | 78±2 |
| SACNNs | **0.9961** | **0.9167** | **0.6938** | **0.7358** | **0.6007** | **0.2913** | 136±2 |

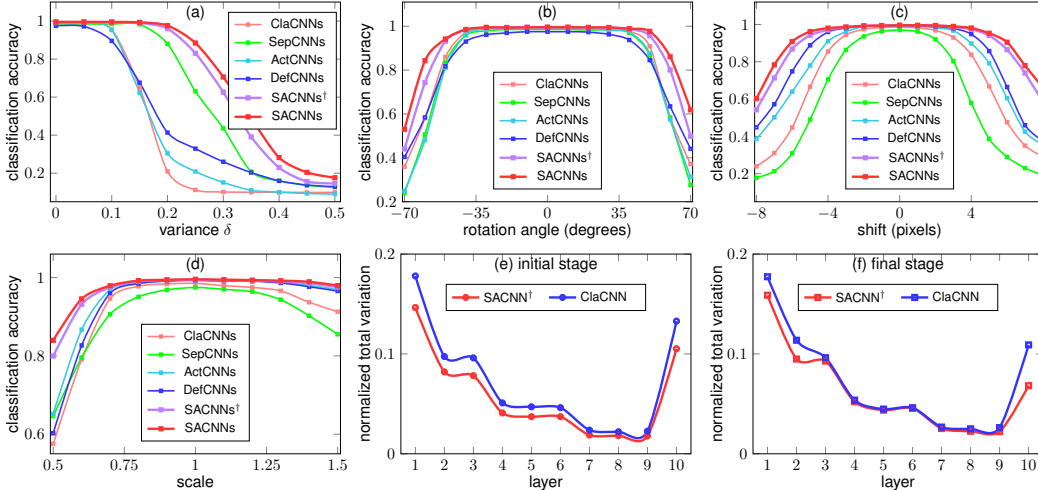

Figure 2: Invariance properties of various CNNs. (a) Gaussion noises with mean 0 and variance $\delta$. (b) Rotation. (c) Shift. (d) Scale. (e) Normalized total variations at the initial stage. (f) Normalized total variations at the final stage. Large figures can be found in the supplementary material.

error loss is used in the regression tasks. During the testing stage, the squared correlation and the root mean square error are used to evaluate the results on DPP4 and TF-198 respectively, and the classification or clustering accuracy is used for the others. For a reasonable evaluation, we perform 5 random restarts and the average results are used for comparisons.

### 5.2 Compared with various CNNs on Euclidean domains

To validate the capability of SACNNs on the Euclidean domains, several SACNNs are modeled to classify images in Mnist, Cifar-10, Cifar-100 and STL-10, and to cluster images in Image10 and ImageDog based on the DAC model [6]. In this experiment, images are recast as specific multi-channel graphs on 2-Dimensional regular grids. In the graphs, each vertex is provided with 9 neighbors including itself, which is similar to the classical convolution with a $3 \times 3$ filter.

In Table 1, we report the quantitative results of the modeled networks with diverse convolution units on various Euclidean structured datasets. Note that SACNNs achieve the superior performance on both classification and clustering tasks, which implies that SACNNs and SACNNs† are capable of managing Euclidean structured data effectively. In Figure 2, we empirically verify the invariance property of the compared CNNs on the Mnist dataset. In this experiment, we disturb the testing data in Mnist with four typical transformations, including Gaussion noise, rotation, shift, and scale. Then, these disturbed data is utilized to validate the trained networks with the evaluated convolution units. From Figure 2, the results assuredly prove that SACNNs and SACNNs† are in possession of excellent robustness to such transformations. Furthermore, we analyze the learned filters via the normalized total variation [33] that can reveal the smoothness of filters. Figure 2 (e) and (f) show that smoother filters are obtained in SACNNs† at both initial and final stages. Based on the conclusion in [33], higher deformation stability will be achieved when smoother filters are learned, which is in agreement with the results of our experiments in Figure 2 (a)-(d).

Table 2: The results on the experimental non-Euclidean structured datasets. For each dataset, $\uparrow (\downarrow)$ indicates that the larger (the smaller) values, the better results are.

| Datasets | Mnist↑ | 20News↑ | Reuters↑ | NTU↑ | DPP4↑ | TF-198↓ | Time (s) |
|---|---|---|---|---|---|---|---|
| LCNs [4] | 0.9914 | 0.6491 | 0.9162 | 0.5457 | 0.225 | 68.83 | 175±2 |
| DFNs [40] | 0.9840 | 0.7017 | 0.9046 | 0.6346 | 0.214 | 70.35 | 192±3 |
| ECC [37] | 0.9937 | 0.7003 | 0.9114 | 0.6416 | 0.249 | 65.35 | 238±4 |
| MoNets [31] | 0.9919 | 0.6929 | 0.9113 | 0.6354 | 0.256 | 69.35 | 252±4 |
| SCNs [16] | 0.9726 | 0.6453 | 0.8985 | 0.5818 | 0.248 | 75.83 | 1384±11 |
| ChebNets [10] | 0.9914 | 0.6826 | 0.9124 | 0.6384 | 0.265 | 65.86 | 673±8 |
| GCNs [22] | 0.9867 | 0.6278 | 0.8992 | 0.5983 | 0.258 | 71.54 | 341±4 |
| SACNNs[†] | 0.9957 | 0.7362 | 0.9365 | 0.6844 | 0.279 | 58.82 | **78±2** |
| SACNNs | **0.9961** | **0.7436** | **0.9452** | **0.6931** | **0.285** | **53.72** | 136±2 |

## 5.3 Compared with diverse GCNNs on non-Euclidean domains

To verify the versatility of SACNNs for non-Euclidean structured data, we build SACNNs to classify the texts in 20News and Reuters, recognize the skeleton-based actions in NTU, estimate the activities of molecules in DPP4, and predict the taxis flows in TF-198, respectively. In addition, Mnist is also used to see how these GCNNs perform on Euclidean structured data.

Table 2 gives the results in this experiment, which shows that SACNNs and SACNNs[†] outperform all the compared methods with significant margins. In addition, we have several observations from the table. First, dramatical improvements are achieved by SACNNs on both Euclidean and non-Euclidean domains in numerous tasks. Such a good performance verifies that SACNNs can effectively deal with data on different domains, without any human intervention. Second, Table 1, Table 2 and Figure 2 consistently show that SACNNs always achieve better performance than SACNNs[†]. These results empirically confirm that the local structure representation learning is capable of capturing the significant structure information from data, thus improving the capability of SACNNs with only a few additional learnable parameters. Furthermore, Table 1 and Table 2 report the time consumptions of the evaluated methods when one epoch is executed on Mnist during training. From these tables, we observe that SACNNs are obviously faster than the competitive GCNN methods. Compared with the CNN methods, the timing cost of SACNNs is tolerably, which ensures the practicability of SACNNs.

## 5.4 Ablation study

In this subsection, we perform extensive ablation studies on diverse datasets to synthetically analyze the developed SACNNs. Intuitively, all the results are illustrated in Figure 3. Due to the space limitation, the learned filters in SACNNs are presented in the supplementary material.

**Impact of polynomial order** To show the impact of polynomial order $t$ on the structure-aware convolution, we select $t$ from $\{5, 40, 80, 120, 160\}$ to generate $11 \times 11$ filters to classify STL-10. Figure 3 (a) illustrates the validation errors of SACNNs with different $t$. One can observe that the performance generally improves if we increase the polynomial order $t$, then the performance will saturate when filters can be well approximated, *i.e.*, $t \geq 80$ is satisfied. Moreover, it is worthy to note that the developed SACNNs can utilize parameters more effectively than ClaCNNs. This is empirically supported by the observation that SACNNs with only 40 parameters per filter can achieve significant better performance than ClaCNNs with $11 \times 11 = 121$ parameters per filter.

**Influence of channels** On the Cifar-10 dataset, we model SACNNs with different numbers of channels $c$ (*i.e.*, 8, 16, 32) to study its influence on the local structure representations learning. Specifically, we observe the following two tendencies from Figure 3 (b). The first one is that the performance of both SACNNs and ClaCNNs benefits from the increase of the channel numbers. This is reasonable since more parameters may improve the expressive capability of networks in general. Second, our SACNNs work consistently better than ClaCNNs, especially when the channel number is relatively large. One considerable reason is that more information can be exploited to model the latent structure information to assist SACNNs achieving superior performance.

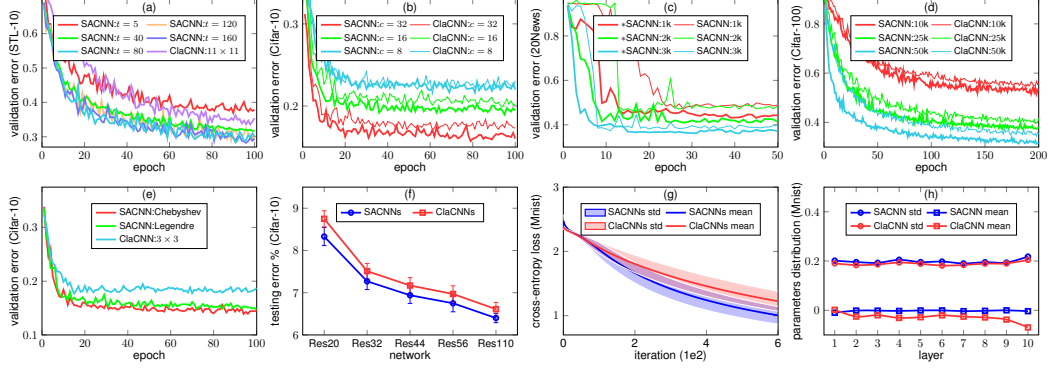

Figure 3: Ablation studies on various datasets. (a) Impact of polynomial order. (b) Influence of channels. (c) Transfer learning from Reuters to 20News. (d) Impact of training samples. (e) Influence of basis functions. (f) Integration with recent networks. (g) Sensitivity to initialization. (h) Parameters distribution. Large figures can be found in the supplementary material.

**Transfer learning from Reuters to 20News**  To reveal the transferability of SACNNs, we fine-tune the SACNNs that are pre-trained on Reuters (denoted as *SACNNs), with a small number of labeled samples (*i.e.*, 1k, 2k, 3k) in the 20News dataset. Figure 3 (c) shows that the pre-training on Reuters can significantly elevate the performance of SACNNs on 20News and stabilize the training process simultaneously, especially when labeled training samples are limited. This demonstrates that SACNNs learned on a domain can be seamlessly transferred to similar domains.

**Impact of training samples**  We randomly sample three sub-datasets with various sizes (*i.e.*, 10k, 25k, 50k) from Cifar-100 to evaluate the impact of number of training samples on SACNNs. As illustrated in Figure 3 (d), the performance of SACNNs improves when more training samples are used. Furthermore, the superiority of our SACNNs against ClaCNNs holds on all these cases, which means that SACNNs are capable of tackling machine learning tasks with both rich and limited data.

**Influence of basis functions**  To investigate the influence of basis functions on SACNNs, the Legendre polynomials are employed as basis functions to learn filters on Cifar-10. Similar to the Chebyshev polynomials, the Legendre polynomial $h_k(x)$ of order $k-1$ ($k \geq 3$) can be obtained based on the recurrence relation $h_k(x) = \frac{2k+1}{k+1}h_{k-1}(x) - \frac{k}{k+1}h_{k-2}(x)$, with $h_1(x) = 1$ and $h_2(x) = x$. From Figure 3 (e), almost the same training processes are generated in spite of diverse bases. The slight mismatching may come from the randomness in training, *e.g.*, random mini-batch selections. This demonstrates that the learnability of SACNNs is robust to the basis functions.

**Integration with recent networks**  A class of popular networks, *i.e.*, ResNets [15], are employed to survey the range of applications of our structure-aware convolution. The results in Figure 3 (f) clearly indicate that better improvements will be achieved by replacing the classical convolution in ResNets with the structure-aware convolution. This adequately validates that the structure-aware convolution suffices to be applied to general ClaCNNs, not confined to simple and shallow networks.

**Sensitivity to initialization**  We carry out an experiment on Mnist to contrastively analyze the sensitivities to initializations in SACNNs and ClaCNNs. In this experiment, parameters in networks are randomly initialized with a uniform distribution $U(-\alpha, \alpha)$, where $\alpha$ is randomly selected from $[0, 1]$. Figure 3 (g) illustrates the descending processes of loss functions in ClaCNNs and SACNNs, indicating that SACNNs generally converge faster than ClaCNNs and are robust to initializations. A possible reason is that the whole values in the generated discrete filters $\mathbf{f}_{\mathcal{R}_i} = \{f(r_{ji})|e_{ji} \in \mathcal{E}\}$ can be together modified by adjusting each coefficient of basis functions, which may yield more precise gradients to accelerate and stabilize the training processes.

**Parameters distribution**  Figure 3 (h) shows the distributions of parameters learned by ClaCNNs and SACNNs on Mnist in ten convolutional layers. From the figure, we have the following two observations. First, the parameters in both SACNNs and ClaCNNs have almost the same standard deviations. Second, the expectations of parameters in SACNNs are more closer to 0 than ClaCNNs.

These observations reveal that SACNNs have more sparse parameters than ClaCNNs. As a result, more robust models will be achieved, which is in accordance with the results in Section 5.2.

## 6  Conclusion

We present a conceptually simple yet powerful structure-aware convolution to establish SACNNs. In the structure-aware convolution, filters are represented via univariate functions, which suffice to aggregate local inputs with diverse topological structures. By feat of the function approximation theory, a numerical strategy is proposed to learn these filters in an effectively and efficiently way. Furthermore, rather than using the predefined local structures of data, we incorporate them into the structure-aware convolution to learn the underlying structure information from data automatically. Extensive experimental results strongly demonstrate that the structure-aware convolution can be equipped in SACNNs to learn high-level representations and latent structures for both Euclidean and non-Euclidean structured data. In the future, we plan to systematically investigate the interpretability of SACNNs based on their functional filters, *i.e.*, univariate functions.

**Acknowledgments**

This work was supported by the National Natural Science Foundation of China under Grants 91646207, 61773377 and 61573352, and the Beijing Natural Science Foundation under Grants L172053. We would like to thank Lele Yu, Bin Fan, Cheng Da, Tingzhao Yu, Xue Ye, Hongfei Xiao, and Qi Zhang for their invaluable contributions in shaping the early stage of this work.

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
