[Supplementary Material]

# Supplementary Material:
# Structure-Aware Convolutional Neural Networks

**Jianlong Chang**[1,2]     **Jie Gu**[1,2]     **Lingfeng Wang**[1]     **Gaofeng Meng**[1]
**Shiming Xiang**[1,2]     **Chunhong Pan**[1]
[1]NLPR, Institute of Automation, Chinese Academy of Sciences
[2]School of Artificial Intelligence, University of Chinese Academy of Sciences
{jianlong.chang, jie.gu, lfwang, gfmeng, smxiang, chpan}@nlpr.ia.ac.cn

## Abstract

This is the supplementary material for the paper entitled "Structure-Aware Convolutional Neural Networks". In this material, Section 1 presents the the proof of Theorem 1. Section 2 clarifies the descriptions related with the experiments, including the gradients in training, experimental datasets, and network architectures.

## 1   Proof of Theorem 1

**Theorem 1.** *Under the Chebyshev polynomial basis, the structure-aware convolution is equivalent to*

$$y_i = \mathbf{v}^{\mathrm{T}} \mathbf{P}_i \mathbf{x}_i, \quad i \in \{1, 2, \cdots, n\},$$

*where $\mathbf{v} \in \mathbb{R}^t$ is the coefficients of the polynomials, $\mathbf{P}_i \in \mathbb{R}^{t \times m}$ is a matrix determined by the local structure representation $\mathcal{R}_i$ and the polynomials, and $\mathbf{x}_i \in \mathbb{R}^m$ is the local input at the $i$-th vertex.*

*Proof.* For clarity, we denote $\mathbf{x} \in \mathbb{R}^m$ and $\mathcal{R}_i = \{r_{ji} | e_{ji} \in \mathcal{E}\}$ as the local input and local structure representations at the $i$-th vertex, and $f(\cdot)$ is an functional filter. In the structure-aware convolution, the output at the $i$-th location (vertex) is formulated as

$$y_i = \sum_{j=1}^{m} f(r_{ji}) \cdot x_j = \sum_{j=1}^{m} \left( \sum_{k=1}^{t} v_k \cdot h_k(r_{ji}) \right) \cdot x_j. \tag{1}$$

By representing the Chebyshev polynomial basis with $\{1, x^1, x^2, \cdots, x^{t-1}\}$, we have

$$\begin{bmatrix} h_1(x) \\ h_2(x) \\ h_3(x) \\ \vdots \\ h_t(x) \end{bmatrix} = \mathbf{T} \begin{bmatrix} 1 \\ x^1 \\ x^2 \\ \vdots \\ x^{t-1} \end{bmatrix}, \tag{2}$$

where $\mathbf{T}$ is purely determined by the Chebyshev polynomial basis.

Then, we have

$$\sum_{k=1}^{t} v_k \cdot h_k(r_{ji}) = [v_1, v_2, v_3, \cdots, v_t] \begin{bmatrix} h_1(r_{ji}) \\ h_2(r_{ji}) \\ h_3(r_{ji}) \\ \vdots \\ h_{t-1}(r_{ji}) \end{bmatrix} = \mathbf{v}^{\mathrm{T}} \mathbf{T} \begin{bmatrix} 1 \\ r_{ji}^1 \\ r_{ji}^2 \\ \vdots \\ r_{ji}^{t-1} \end{bmatrix}. \tag{3}$$

According to Eq. (2) and Eq. (3), we have

$$
\begin{aligned}
y_i &= \mathbf{v}^{\mathrm{T}} \mathbf{T}
\begin{pmatrix}
1 & 1 & 1 & \cdots & 1 \\
r_{1i}^1 & r_{2i}^1 & r_{3i}^1 & \cdots & r_{mi}^1 \\
r_{1i}^2 & r_{2i}^2 & r_{3i}^2 & \cdots & r_{mi}^2 \\
\vdots & \vdots & \vdots & \ddots & \vdots \\
r_{1i}^{t-1} & r_{2i}^{t-1} & r_{3i}^{t-1} & \cdots & r_{mi}^{t-1}
\end{pmatrix}
\begin{bmatrix}
x_1 \\
x_2 \\
x_3 \\
\vdots \\
x_m
\end{bmatrix}, \\
&= \mathbf{v}^{\mathrm{T}} \mathbf{T} \mathbf{G}_i \mathbf{x}_i, \\
&= \mathbf{v}^{\mathrm{T}} \mathbf{P}_i \mathbf{x}_i,
\end{aligned}
\tag{4}
$$

where $\mathbf{T}$ and $\mathbf{G}_i$ are determined by the basis and the local structure representation $\mathcal{R}_i$, respectively. The proof is completed. $\qquad\square$

## 2 Experiments

### 2.1 Training SACNNs with back-propagation

During training, we need to compute the gradients with respect to the parameters in the structure-aware convolution, *i.e.*, $\mathbf{v}$ and $\mathbf{M}$. For clarity, we formulate the gradients in a structure-aware convolutional layer with the multi-channel input $\mathbf{x} \in \mathbb{R}^{n \times c}$ and the single-channel output $\mathbf{y} \in \mathbb{R}^n$. During the forward propagation phase, $\mathbf{y}$ can be computed as

$$
y_i = \sum_{u=1}^{c} \sum_{e_{ji} \in \mathcal{E}} f_u(r_{ji}) \cdot x_{ju} = \sum_{u=1}^{c} \sum_{e_{ji} \in \mathcal{E}} \left( \sum_{k=1}^{t} v_{ku} \cdot h_k(r_{ji}) \right) \cdot x_{ju}, \quad i \in \{1, 2, \cdots, n\}.
\tag{5}
$$

During the backward propagation phase, the gradients can be computed as

$$
\begin{aligned}
\frac{\partial y_i}{\partial v_{ku}} &= \sum_{e_{ji} \in \mathcal{E}} x_{ju} \cdot h_k(r_{ji}), \\
\frac{\partial y_i}{\partial M_{pq}} &= \sum_{u=1}^{c} \sum_{e_{ji} \in \mathcal{E}} x_{ju} \cdot \sum_{k=1}^{t} \left( v_{ku} \cdot \frac{\partial h_k}{\partial r_{ji}} \cdot \left(1 - r_{ji}^2\right) \cdot x_{jp} \cdot x_{iq} \right).
\end{aligned}
\tag{6}
$$

Thus, the operations in structure-aware convolution are differentiable, enabling the end-to-end training of the established SACNNs with the standard back-propagation, without additional modification.

### 2.2 Datasets

We perform extensive experiments on six Euclidean and five non-Euclidean structured datasets. The details of datasets are listed in Table 1. In each experiment, specifically, the whole samples are normalized by subtracting the mean values in each dimension. Note that the data augmentation technique is omitted in the ablation studies to reduce the impacts of additional factors.

The Euclidean structured datasets include six image datasets, namely Mnist [13], Cifar-10 [11], Cifar-100 [11], STL-10 [4], Image10 [2], and ImageDog [2]. In the image classification tasks, Mnist, Cifar-10, Cifar-100, and STL-10 are utilized to test the capability of classification of SACNNs. In the image clustering tasks, DAC [2] modeled with our SACNNs is validated on the Image10 [2] and ImageDog [2] datasets, which are sampled from ImageNet [6].

For the non-Euclidean structured datasets, five datasets are employed, namely the text categorization datasets 20NEWS and Reuters [12], the action recognition dataset NTU [14], the molecular activity dataset DPP4 [10], and the taxi flow dataset TF-198. The 20News and Reuters datasets are collected from $16,381$ and $9,160$ text documents associated with 20 classes. In experiments, the bag-of-words model [1] is employed to encode each document as a graph. For each sample in the NTU dataset, each vertex represents a part of body, edges indicate the correlations between vertices [5]. In DPP4, the target is to calculate activities of molecules based on the molecule structures, which can be

Table 1: Descriptions of experimental datasets. For clarity, "classifi.", "cluster.", "regress.", "mse", "corr." are the abbreviations of classification, clustering, regression, mean square error, and correlation respectively. Note that "#samples (Tr)" and "#samples (Te)" denote the numbers of training and testing samples respectively, "#edges" means the average number of edges of each samples in datasets, $+\infty$ signifies that the outputs are continuous.

| Euclidean structured datasets | | | | | | |
|---|---|---|---|---|---|---|
| Datasets | Mnist | Cifar-10 | Cifar-100 | STL-10 | Image10 | ImageDog |
| #problem | classifi. | classifi. | classifi. | classifi. | cluster. | cluster. |
| #classes | 10 | 10 | 100 | 10 | 10 | 15 |
| #samples (Tr) | 60,000 | 50,000 | 50,000 | 8,000 | 13,000 | 19,500 |
| #samples (Te) | 10,000 | 10,000 | 10,000 | 5,000 | 13,000 | 19,500 |
| #features | $28 \times 28$ | —— $32 \times 32 \times 3$ —— | | —————— $96 \times 96 \times 3$ —————— | | |
| #losses | ———————————— categorical cross entropy ———————————— | | | | | |
| #evaluation | ———————— classification/clustering accuracy ———————— | | | | | |

| non-Euclidean structured datasets | | | | | | |
|---|---|---|---|---|---|---|
| Datasets | Mnist | 20News | Reuters | NTU | DPP4 | TF-198 |
| #problem | classifi. | classifi. | classifi. | classifi. | regress. | regress. |
| #classes | 10 | 20 | 20 | 60 | $+\infty$ | $+\infty$ |
| #samples (Tr) | 60,000 | 10,904 | 6,577 | 33,628 | 6,148 | 20,000 |
| #samples (Te) | 10,000 | 5,477 | 2,583 | 11,209 | 2,045 | 9,981 |
| #nodes | 784 | 19,381 | 1,500 | 25 | 2,153 | 198 |
| #edges | 5940 | 73,732 | 13,862 | 45 | 17,767 | 1,586 |
| #losses | ——— categorical cross entropy ——— | | | | mse | mse |
| #evaluation | ———— classification accuracy ———— | | | | squared corr. | root mse |

considered a regression problem on the non-Euclidean domain. The TF-198 dataset [16] is collected from the GPS trajectory data of $29,950$ taxis in a city from November 2015 to May 2016 at $198$ traffic intersections. For each sample, six history observations (120 minutes) and the taxis flow of next time interval (20 minutes) are acted as inputs and outputs, respectively.

## 2.3 Networks modeling

Our code relies on Keras [3] with the Tensorflow [1] backend. Basically, the max pooling and the Graclus method [7] are employed as the pooling operations to coarsen the feature maps in SACNNs when managing Euclidean and non-Euclidean structured data respectively, the ReLU function [8] is used as the activation function, batch normalization [9] is employed to normalize the inputs of all layers, parameters are randomly initialized with a uniform distribution $U(-0.1, 0.1)$, the small filters with $3 \times 3$ size are always utilized in the convolutional layers because of their capability of nonlinear fitting, as verified in [15]. In the experiments, the architectures are fixed, only the convolution units are different. For each experimental dataset, the devised networks are listed in Table 2. In the table, we abbreviate units in networks for clarity. Specifically, the convolutional layers with BN and ReLU are denoted as "[kernel size] conv. [number of channels] BN ReLU", the fully connected layers with BN and ReLU are expressed as "[output dimension] fc BN ReLU", the max pooling layers with BN are presented as "[pooling size] max-pooling BN", and the global averaging pooling layers with BN are signified as "[global averaging pooling size] global averaging BN".

## 2.4 Experimental results

We report the high-definition large figures of the results in the paper. Specifically, the results of invariance properties in Section 5.2 are illustrated in Figure 1, the results in Section 5.4 are shown in Figure 2, and the learned filters are presented in Figure 3.

**Filters visualization** To validate whether SACNNs can learn interpretable filters, we model a new network for STL-10 by replacing the first three convolutional layers in the network (in Table 2) for STL-10 with a convolutional layer in which the filer size is $11 \times 11$ pixels. In Figure 3, the learned filters ($11 \times 11$ pixels) in the first convolutional layer are intuitively presented, which shows that both SACNNs and ClaCNNs tend to learn a variety of frequency and orientation selective filters.

Table 2: The architectures of the modeled networks on the experimental datasets (shown in columns).

| | Euclidean structured datasets | | |
|---|---|---|---|
| Dataset | Mnist | Cifar-10 | Cifar-100 |
| Network | $3 \times 3$ conv. 32 BN ReLU<br>$3 \times 3$ conv. 32 BN ReLU<br>$3 \times 3$ conv. 32 BN ReLU<br>$2 \times 2$ max-pooling BN<br>$3 \times 3$ conv. 64 BN ReLU<br>$3 \times 3$ conv. 64 BN ReLU<br>$3 \times 3$ conv. 64 BN ReLU<br>$2 \times 2$ max-pooling BN<br>$3 \times 3$ conv. 128 BN ReLU<br>$3 \times 3$ conv. 128 BN ReLU<br>$3 \times 3$ conv. 128 BN ReLU<br>$2 \times 2$ max-pooling BN<br>$3 \times 3$ conv. 10 BN ReLU<br>$7 \times 7$ global averaging BN<br>10 softmax | $3 \times 3$ conv. 32 BN ReLU<br>$3 \times 3$ conv. 32 BN ReLU<br>$3 \times 3$ conv. 32 BN ReLU<br>$2 \times 2$ max-pooling BN<br>$3 \times 3$ conv. 64 BN ReLU<br>$3 \times 3$ conv. 64 BN ReLU<br>$3 \times 3$ conv. 64 BN ReLU<br>$2 \times 2$ max-pooling BN<br>$3 \times 3$ conv. 128 BN ReLU<br>$3 \times 3$ conv. 128 BN ReLU<br>$3 \times 3$ conv. 128 BN ReLU<br>$2 \times 2$ max-pooling BN<br>fc 1024 BN ReLU<br>fc 512 BN ReLU<br>10 softmax | $3 \times 3$ conv. 32 BN ReLU<br>$3 \times 3$ conv. 32 BN ReLU<br>$3 \times 3$ conv. 32 BN ReLU<br>$2 \times 2$ max-pooling BN<br>$3 \times 3$ conv. 64 BN ReLU<br>$3 \times 3$ conv. 64 BN ReLU<br>$3 \times 3$ conv. 64 BN ReLU<br>$2 \times 2$ max-pooling BN<br>$3 \times 3$ conv. 128 BN ReLU<br>$3 \times 3$ conv. 128 BN ReLU<br>$3 \times 3$ conv. 128 BN ReLU<br>$2 \times 2$ max-pooling BN<br>fc 1024 BN ReLU<br>fc 512 BN ReLU<br>100 softmax |
| Dataset | STL-10 | Image10‡ | ImageDog‡ |
| Network | $3 \times 3$ conv. 32 BN ReLU<br>$3 \times 3$ conv. 32 BN ReLU<br>$3 \times 3$ conv. 32 BN ReLU<br>$2 \times 2$ max-pooling BN<br>$3 \times 3$ conv. 32 BN ReLU<br>$3 \times 3$ conv. 32 BN ReLU<br>$3 \times 3$ conv. 32 BN ReLU<br>$2 \times 2$ max-pooling BN<br>$3 \times 3$ conv. 32 BN ReLU<br>$3 \times 3$ conv. 32 BN ReLU<br>$3 \times 3$ conv. 32 BN ReLU<br>$2 \times 2$ max-pooling BN<br>$3 \times 3$ conv. 32 BN ReLU<br>$3 \times 3$ conv. 32 BN ReLU<br>$3 \times 3$ conv. 32 BN ReLU<br>$2 \times 2$ max-pooling BN<br>fc 1024 BN ReLU<br>fc 1024 BN ReLU<br>10 softmax | $3 \times 3$ conv. 64 BN ReLU<br>$3 \times 3$ conv. 64 BN ReLU<br>$3 \times 3$ conv. 64 BN ReLU<br>$2 \times 2$ max-pooling BN<br>$3 \times 3$ conv. 128 BN ReLU<br>$3 \times 3$ conv. 128 BN ReLU<br>$3 \times 3$ conv. 128 BN ReLU<br>$2 \times 2$ max-pooling BN<br>$3 \times 3$ conv. 256 BN ReLU<br>$3 \times 3$ conv. 256 BN ReLU<br>$3 \times 3$ conv. 256 BN ReLU<br>$2 \times 2$ max-pooling BN<br>$1 \times 1$ conv. 10 BN ReLU<br>$6 \times 6$ global averaging BN<br>10 fc BN ReLU<br>10 fc BN ReLU<br>restraint layer | $3 \times 3$ conv. 64 BN ReLU<br>$3 \times 3$ conv. 64 BN ReLU<br>$3 \times 3$ conv. 64 BN ReLU<br>$2 \times 2$ max-pooling BN<br>$3 \times 3$ conv. 128 BN ReLU<br>$3 \times 3$ conv. 128 BN ReLU<br>$3 \times 3$ conv. 128 BN ReLU<br>$2 \times 2$ max-pooling BN<br>$3 \times 3$ conv. 256 BN ReLU<br>$3 \times 3$ conv. 256 BN ReLU<br>$3 \times 3$ conv. 256 BN ReLU<br>$2 \times 2$ max-pooling BN<br>$1 \times 1$ conv. 15 BN ReLU<br>$6 \times 6$ global averaging BN<br>15 fc BN ReLU<br>15 fc BN ReLU<br>restraint layer |

| | non-Euclidean structured datasets | | |
|---|---|---|---|
| Dataset | 20News | Reuters | NTU |
| Network | conv. 16 BN ReLU<br>conv. 32 BN ReLU<br>fc 512 BN ReLU<br>20 softmax | conv. 16 BN ReLU<br>conv. 32 BN ReLU<br>fc 512 BN ReLU<br>20 softmax | conv. 32 BN ReLU<br>conv. 32 BN ReLU<br>conv. 32 BN ReLU<br>fc 1024 BN ReLU<br>60 softmax |
| Dataset | DPP4 | TF-198 | |
| Network | conv. 8 BN ReLU<br>conv. 16 BN ReLU<br>conv. 32 BN ReLU<br>fc 512 BN ReLU<br>1 ReLU | conv. 32 BN ReLU<br>conv. 32 BN ReLU<br>conv. 32 BN ReLU<br>conv. 1 ReLU | |

Figure 1: Invariance properties of various CNNs. (a) Gaussion noises with mean 0 and variance $\delta$. (b) Rotation. (c) Shift. (d) Scale. (e) Normalized total variations at the initial stage. (f) Normalized total variations at the final stage.

Figure 2: Ablation studies on various datasets. (a) Impact of polynomial order. (b) Influence of channels. (c) Transfer learning from Reuters to 20News. (d) Impact of training samples. (e) Influence of basis functions. (f) Integration with recent networks. (g) Sensitivity to initialization. (h) Parameters distribution.

Figure 3: Visualization of learned filters ($11 \times 11 \times 32$) on the STL-10 dataset. (a) The learned classical filters. (b) The discrete filters generated from the learned functional filters. (c) The learned functional filters. The red, green and blue curves respectively correspond to the red, green and blue channels of samples in STL-10.

## Footnotes

[1] https://pypi.python.org/pypi/bagofwords