[Reviews · NeurIPS 2018]

Reviewer 1



The paper develops a structure-aware convolution neural network to cope with both Euclidean and non-Euclidean structured data. Strengths: 1) The proposed structure-aware convolution neural network is capable of dealing with both Euclidean and non-Euclidean structured data. 2) It is easy to understand the paper. Weaknesses: 1) The technical depth of the paper is shallow. It lacks deep theoretical underpinning. The key idea is mainly contained in Eq. (5), which is basically kernel approximation using truncated polynomials. The idea is simple and straightforward. It does not represent a major breakthrough. 2) The paper mainly conducts empirical studies by applying the proposed structure-aware convolution neural network to various data sets. But the performance gain is insignificant and the cost is high.

Reviewer 2



This paper proposes a good idea to explicitly utilize the input data structures in CNN (such as non-Euclidean or graph structures), and the proposed model shows the superior performance for different tasks including vision and text. The idea is intuitive and reasonable, and learning functional filters and the representations is sound. The presentation of this paper is clear and detailed. Moreover, the conducted experiments contain several tasks, and the proposed model achieve better results consistently. The analysis and discussion are informative and can serve the guidance for future research work. The code will be released so that the model can be easily reproduced to benefit following work. After seeing author responses, I kept my positive comments but made a small change of the final score.

Reviewer 3



The paper proposes a structure-aware convolution structure that can naturally aggregate local inputs from diverse topological structures without predefining data structures. The learnable filters in the convolution operation are well designed in a simple way. The methods shows better quantitative results on eleven datasets when comparing to baselines. Strength: 1. Simple. The convolution operator is simply unified on diverse topological structures. 2. Extensive experiments on six Euclidean and five non-Euclidean structured datasets. 3. Results show the proposed method has certain improvements over recent convolution neural network and graph convolution networks. 4. The paper is well written, and clear to my understanding. Weakness: 1. Notation confusion. In equation (1) and theorem #1, x_i denotes a local input from (i-m)-th vertex to (i+m)-th vertex. However, in equation (6), a same x_i represents a single value (with multi-channel) at i-th vertex only. 2. The detailed information for graph construction is missing, e.g. the way to define edges and construct a graph for non-euclidean datasets. 3. Despite the proposed method shares a different idea with [17], the P_i matrix in theorem #1 mathematically transforms the input x_i into a high-dimensional space where a convolution weights v are applied. Can the authors briefly compares the proposed method with [17]? My feeling is that [17] should be a good baseline to compare on Euclidean datasets. 4. It seems that the baselines in Experiment 5.2 are not the original models, but the authors' own implementations, e.g. ActCNNs[18] has 93% accuracy on Cifar10 instead of 90%. Can the authors comment about this? 5. It would be good if the author can compare number of parameters between the proposed network and used baselines. Minor comments: 1. In figure 3(f), the improvement looks saturated when using 56 layers network. Will the proposed method also benefit deeper network (e.g. resnet110 in Cifar10)? 2. It would be interesting if the authors would comment on how their approach can be applied to point cloud dataset. Conclusion: I hope the authors will response the above questions in the rebuttal and include these in the final version. Given the novelty of the approach and extensive experiments, the submission would be stronger if the authors could fix these flaws.